# Sex: A Significant Risk Factor for Neurodevelopmental and Neurodegenerative Disorders

**DOI:** 10.3390/brainsci8080154

**Published:** 2018-08-13

**Authors:** Paulo Pinares-Garcia, Marielle Stratikopoulos, Alice Zagato, Hannah Loke, Joohyung Lee

**Affiliations:** 1Brain and Gender laboratory, Centre for Endocrinology and Metabolism, Hudson Institute of Medical Research, Clayton, Victoria 3168, Australia; paulo.pinares-garcia@hudson.org.au (P.P.-G.); mstr0005@student.monash.edu (M.S.); alice.zagato@hudson.org.au (A.Z.); hannah.loke@hudson.org.au (H.L.); 2Department of Anatomy and Developmental Biology, Monash University, Clayton, Victoria 3168, Australia; 3School of Life and Environmental Sciences, Deakin University, Burwood, Victoria 3125, Australia

**Keywords:** brain sex differences, estrogen, testosterone, *SRY*, gender-specific medicine, ADHD, Parkinson’s disease, Alzheimer’s disease, autism, schizophrenia, depression

## Abstract

Males and females sometimes significantly differ in their propensity to develop neurological disorders. Females suffer more from mood disorders such as depression and anxiety, whereas males are more susceptible to deficits in the dopamine system including Parkinson’s disease (PD), attention-deficit hyperactivity disorder (ADHD) and autism. Despite this, biological sex is rarely considered when making treatment decisions in neurological disorders. A better understanding of the molecular mechanism(s) underlying sex differences in the healthy and diseased brain will help to devise diagnostic and therapeutic strategies optimal for each sex. Thus, the aim of this review is to discuss the available evidence on sex differences in neuropsychiatric and neurodegenerative disorders regarding prevalence, progression, symptoms and response to therapy. We also discuss the sex-related factors such as gonadal sex hormones and sex chromosome genes and how these might help to explain some of the clinically observed sex differences in these disorders. In particular, we highlight the emerging role of the Y-chromosome gene, *SRY,* in the male brain and its potential role as a male-specific risk factor for disorders such as PD, autism, and ADHD in many individuals.

## 1. Introduction

Sex is sometimes a significant variable in the prevalence and incidence of neurological disorders [1,2,3,4,5,6,7]. In addition, sexual differences also exist in the age-of-onset, progression, disease severity, underlying neuropathology and treatment response of neurological diseases [1,3,4,5,7,8,9]. Whilst this attention to sexual dimorphism has traditionally been stronger in fields like cancer, cardiovascular and endocrine disorders [10,11], accumulating evidence demonstrates significant sex differences in brain physiology and behavior throughout development and adulthood [1,4,12]. Indeed, girls are more likely to suffer from depression than boys, and following puberty this female susceptibility for anxiety increases to around twice that for males [13]. Males with ADHD tend to exhibit the hyperactive-impulsive subtype whereas females tend to exhibit the inattentive subtype [14]. Sex differences are also seen in neurodegenerative disorders as the male sex is a significant risk factor for Parkinson’s disease [15,16,17] and motor neuron disease [18,19], whilst females are more susceptible to Alzheimer’s disease [20,21] and multiple sclerosis [22,23] (Summarised in Figure 1). Despite the well-established understanding that males and females differ in their predisposition to neurological diseases, gender is rarely considered when making diagnostic or treatment decisions. Hence, a better understanding of the molecular underpinnings behind these sex differences could help develop more targeted therapies with higher success rates, especially in diseases where sex differences are most prominent.

Significant increase in research efforts is now starting to unravel the biological mechanisms responsible for promoting sex-specific characteristics in the healthy and diseased brain [7,24,25,26,27]. It is now well established that organizational and activational effects of sex steroid hormones play an important role in brain sex differentiation [7,28,29,30,31]. Indeed, neuroprotective actions of estrogen in females underlie sex differences in susceptibility to disorders such as PD and schizophrenia [7,32,33,34], whilst aberrant levels of fetal testosterone have been associated with male preponderance to neurodevelopmental disorders such as autism and ADHD [35,36]. In addition to the influence of gonadal hormones, accumulating evidence demonstrate that sex chromosome genes (i.e., X and Y-linked) can directly influence brain function in normal and pathological conditions [26,37,38]. The Y-chromosome gene, *SRY*, which directly exerts male-specific actions in adult dopamine neurons, may also underlie male preponderance to disorders such as PD and autism. Thus, it is likely that these sex differences in the healthy and diseased brain result from complex interactions between sex hormones, sex chromosomes and epigenetic factors.

In view of these findings, increasing our understanding regarding the molecular basis of sex differences in neurological disorders will be pivotal in identifying sex-specific risk or protective factors and in being able to develop more effective therapies for each sex. The current review will summarise the available evidence on sex differences in neuropsychiatric and neurodegenerative disorders regarding prevalence, progression, symptoms and response to therapy. We will then discuss the potential role of sex hormones and sex chromosome genes in contributing to these sex differences. 

## 2. Female-Biased Brain Disorders

### 2.1. Female-Biased Neuropsychiatric Disorders

#### 2.1.1. Anxiety Disorders

Anxiety disorders are a group of psychiatric disorders characterized by exaggerated feelings of anxiety and fear responses. These feelings can manifest as physical symptoms, such as increased heart rate, difficulty concentrating or mind going blank, irritability, muscle tension, lethargy, and sleep disturbance [39,40]. Most common types of anxiety disorders include generalized anxiety disorder, social anxiety disorder, post-traumatic stress disorder, obsessive compulsive disorder and panic disorder. People often have more than one type of anxiety disorder. Like depression, serotonin appears to be involved in the pathogenesis of anxiety disorders and thus therapeutics directed at enhancing synaptic levels of serotonin, such as serotonin-reuptake inhibitors (SSRIs) and serotonin/norepinephrine-reuptake inhibitors (SNRIs), ease symptoms associated with anxiety [41].

Almost all subtypes of anxiety are more likely to be diagnosed in women than in men [42,43]. Lifetime prevalence rates of the major anxiety disorders range between approximately 3 to 12% and are approximately two times greater among women than men [42]. Women seem to be more negatively affected by symptoms of anxiety disorders, often experiencing symptoms to a greater degree [42]. Data on the prevalence of affective disorders such as anxiety and depression mainly come from self-reports and it has been hypothesised that part of the high female prevalence may be due to the unwillingness for males to report their anxiety symptoms [43,44]. Another reason for the higher female prevalence of anxiety disorders has been thought to come from the difficulty men face in expressing their feelings due to masculine sex-role stereotypes [45]. Despite these sociological factors, overwhelming evidence demonstrate that actions of gonadal hormones on brain functions, such as hippocampal neurogenesis [46] and fear conditioning [47], play a significant role in the vulnerability of females to anxiety disorders. 

#### 2.1.2. Depression

Depression is one of the most common and debilitating psychiatric disorders, with more than 300 million people affected worldwide [48]. Depression is a leading cause of suicide [49] characterized by symptoms of depressed mood, lack of drive, anhedonia, changes in appetite, sleep disturbances, feelings of guilt and concentration problems [50,51,52]. These symptoms are thought to largely arise from underactivation of the brain serotonin and norepinephrine transmitter systems [53], although deficiencies in neurotrophic and angiogenic factors [54] and glutamate metabolism [55] are also thought to contribute. Hence, anti-depressant medications such as SSRIs, SNRIs, and monoamine oxidase inhibitors act via enhancing serotonin and/or norepinephrine levels in the brain [56].

Depression is much more common among women than men, with female/male risk ratios roughly 2:1 [57]. This female-bias begins in adolescence and continues to midlife, approximating the span of the childbearing years in women [58]. Like anxiety disorders, sex differences in reporting of symptoms and differential persistence (e.g., sex-roles that create higher stress levels for women leading to higher rates of depression) may contribute to the higher female prevalence of depression [59]. However, increased incidence of depression in women during perimenopause and menopause, as well as following child-birth (i.e., post-partum depression) [43,44,60] suggests that fluctuating levels of sex hormones plays a significant role in female susceptibility to depression.

#### 2.1.3. Late-Onset Schizophrenia 

Schizophrenia is a complex, chronic neuropsychiatric disorder that has a population frequency of approximately 1% [61,62,63]. Schizophrenia is characterized by distortions in thinking, perception, emotions, language, sense of self and behavior, manifested by a mixture of debilitating positive (hallucinations and delusions) and negative symptoms (depression, cognitive impairment, social withdrawal) [64]. Whilst the neuropathology underlying schizophrenia is unclear, most theories center on either an excess or a deficiency of neurotransmitters, including dopamine, serotonin, and glutamate [65,66,67]. Positive symptoms such as hallucinations and delusions, which are thought to result from excessive dopamine release in the prefrontal cortex [68,69] have been treated by dopamine antagonists such as haloperidol [70,71]. Other theories implicate aspartate, glycine, and gamma-aminobutyric acid (GABA) as part of the neurochemical imbalance of schizophrenia [72]. 

Late-onset schizophrenia, which classifies individuals who are diagnosed at the age of 45 or older, are more common in women than men [73]. In contrast, early-onset schizophrenia, which refers to those diagnosed before the age of 18, are more common in males than females (discussed later in this review). Women have been shown to be more vulnerable to psychotic breakdown at times of estrogen withdrawal, i.e., menopause or just after giving birth [74]. Females more frequently exhibit depressive symptoms [73], whilst males tend to have a greater vulnerability to negative symptoms and traits of disorganization. However, studies have found premorbid functioning to be worse in men than in women, with late-onset schizophrenia generally found to have good premorbid competence [75,76,77]. Overall, the prognosis of the illness, the social functioning, and the response to treatment is generally better in female schizophrenic patients compared to males [73]. 

### 2.2. Female-Biased Neurodegenerative Disorders

#### 2.2.1. Alzheimer’s Disease

Alzheimer’s disease (AD) is the most common neurodegenerative disorder, with an estimated worldwide prevalence of 24 million people [78]. AD sufferers exhibit gradual cognitive decline, often beginning with memory loss and extending to behavioural disturbances such as apathy and depression [79]. AD is characterized by the accumulation of intracellular hyperphosphorylated tau inclusions termed neurofibrillary tangles, extracellular plaques consisting of beta-amyloid aggregates, and brain atrophy caused by the progressive loss of cholinergic neurons [80,81,82,83,84]. The propagation of neuropathological changes in AD is highly characterized, beginning in the entorhinal cortex, before spreading to the hippocampus, basal forebrain, temporal and parietal lobes, and eventually the neocortex [85,86]. 

The female sex is a significant risk factor for AD, as two-third of AD patients are women [21,87,88]. Moreover, women have a two-fold higher incidence in AD [89], and two-fold higher lifetime risk of AD compared to men [90,91]. This female-bias is not seen in any other types of dementias [92], although some have reasoned that it is a result of women living longer than men [93]. Women AD patients exhibit faster rate of hippocampal atrophy and greater neurofibrillary tangles than men with AD [94,95]. Women also show more rapid loss of autonomy, greater disability and more rapid cognitive decline whilst men have a higher mortality and comorbidity and later onset [79,96,97]. In particular, female AD patients exhibit a greater cognitive decline in areas of visuospatial abilities, verbal processing, and semantic and episodic memory than male AD patients [98,99,100,101]. Clinical and pre-clinical studies demonstrate conflicting evidence on sex differences in response to anticholinesterase medications, a widely used therapy for AD. Some studies have reported that females responded better than males in clinical [102] and in AD models [103], whilst recent meta-analysis study have no significant sex differences [104]. Overall, women show a greater rate of prevalence and incidence, increased cognitive decline and greater rate of neuropathological decline in AD compared to men. Given the differences in sex hormone levels in AD patients compared to controls, this suggests that biological factors could contribute to sex differences seen in AD. Further investigation into the underlying causes of sex differences in AD is a crucial step towards providing specific gender-based therapies and diagnosis. 

#### 2.2.2. Multiple Sclerosis

Multiple Sclerosis (MS) is a neurodegenerative disease characterized by autoimmune demyelination of axons and plaque formation, eventually leading to CNS degeneration [105]. Disease course can be classified into 4 subtypes; relapsing-remitting (RR), primary progressive (PP), secondary progressive (SP), and primary relapsing (PR). RR-MS as the name suggests, is characterized by periods of acute disease followed by recovery [106]. RR-MS is the most common type accounting for 85–90% of cases, whilst 10–15% of cases are categorized as PP, where symptoms continually worsen with time [106]. The disease mainly affects young adults with peak symptom onset at 30 years of age [107]. MS symptoms have a wide range and severity, resulting from disparity in lesion development and progression. Most commonly reported are fatigue, visual or sensory impairment and cognitive deficits. Magnetic resonance imaging (MRI) is an essential technique for diagnosis and can be used to determine the presence of lesions, anatomically specific to MS whilst brain atrophy is associated with disease progression and cognitive impairment [107,108]. Unlike most neurodegenerative diseases, disease-modifying treatments are available for MS, although there is still no cure. Disease-modifying treatments, which are usually drugs that suppress or modulate the immune systems, are efficacious at slowing disease progression or reducing severity or frequency of attacks when implemented early [109].

There is a higher incidence of MS in females compared to males [22], with 3.6 females to 2.0 male cases per 100,000 [23]. RR-MS has approximately two-fold higher incidence in women over men [105]. Unaffected females that carry susceptibility genes are also more likely to transmit to children than males [110]. RR-MS, female sex and early age of onset are all associated with more benign disease course [111] whilst male sex, later age of onset and a high number of early attacks all associated with poorer prognosis [112,113]. Males with MS are prone to develop less inflammatory, but more destructive lesions than women [114]. Women have greater T-cell immunoreactivity to myelin protein than males in both MS cases and healthy controls [115]. Comparison of gene expression in inflammatory lesions suggests MS pathogenesis in males induces estrogen signalling pathways, whilst females exhibit an upregulation of progesterone pathways [116]. Interestingly, female disease incidence is rising, which may be due to an unknown environmental influence or gene-environment interaction [117,118,119,120,121,122,123].

## 3. Male-Biased Disorders

### 3.1. Male-Biased Neuropsychiatric Disorders

#### 3.1.1. Autism 

Autism, or autism spectrum disorders, is a set of heterogeneous neurodevelopmental conditions, characterized by early-onset difficulties in social communication and unusually restricted, repetitive behaviour and interests [124,125]. Symptoms manifest prior to the age of 3, although diagnosis doesn’t usually occur until 3–4 years of age [126] which is based primarily on the observation of behavioral problems and atypical language development [127]. Globally, autism is thought to affect around 25 million people as of 2015 [128]. The most recent report by the Centers for Disease Control and Prevention revealed a 15% increase in prevalence of autism in the United States from the previous two years [129]. The recent increase in the diagnosis of autism may be partly due to changes in diagnostic practice, although the question of whether actual rates have increased remains unclear [130]. Autism is thought to affect information processing in the brain by altering neuronal connectivity and organization during development [124,125], although how this occurs is not well understood. Evidence from clinical and animal research also suggests an imbalance in serotonin and dopamine [124,125]. Boys with autism have reduced serotonin synthesis and levels in the frontal cortex and thalamus [131,132], which may underlie impaired language production and sensory integration symptoms [131]. Studies have also shown that autism is a hyperdopaminergic condition, likely due to the atypical neural network between the amygdala and prefrontal cortex, which could underlie the social deficits in autism [133,134]. 

Whilst the cause of autism remains unknown, it is likely to result from a combination of genetic and environmental factors such as prenatal infections, valproic acid or alcohol use during pregnancy [124,125]. Interestingly, one of the most consistent findings in autism research is the higher rate of diagnosis in males than females [124,125,135,136]. Autism is approximately four times more common among males than females [137], although this ratio is further increased to eleven males to one female in severe autism [138]. Aside from prevalence rates, males and females differ in the presentation of clinical symptoms [139]. Females with autism show less restricted and repetitive behaviours and interests compared to males and tend to have internalising symptoms such as depression and anxiety, whereas males tend to have more externalising symptoms such as aggression, and hyperactivity [140,141]. Several theories exist on the male preponderance of autism—such as a genetic protective effect in females [142,143] or the “extreme male brain” theory [144]—which will be discussed later in this review. 

#### 3.1.2. Attention-Deficit Hyperactivity Disorder

Attention-deficit hyperactivity disorder (ADHD) is the most commonly diagnosed psychiatric disorder in children, affecting approximately 5% of children worldwide, particularly boys [145]. ADHD is characterized by symptoms of inattention and/or hyperactivity-impulsivity [146]. Symptoms typically appear the age of twelve years old, persist for more than six months, and cause disruptions in at least two settings (such as school, home, or recreational activities). Symptoms of ADHD have been primarily associated with the hypofunction of catecholamines dopamine and norepinephrine in the frontal-subcortical circuit (i.e., prefrontal cortex and striatum), which are involved in attention, reward and motor activity [146,147,148,149]. Thus, the most effective drugs used to treat ADHD, such as methylphenidate and d-amphetamine, are all stimulants, which increase levels of dopamine and norepinephrine to enhance catecholamine signalling in the brain [150,151].

Like autism, ADHD is a neurodevelopmental disorder with a strong male-bias, with a sex ratio of three males to every female [152,153,154,155,156]. Males are likely to exhibit all subtypes of ADHD [157], have higher ADHD symptom scores, and may present with externalising behaviours such as physical abuse, aggression and criminality [158]. On the other hand, females tend to exhibit the inattentive subtype and be at increased risk of developing co-morbid eating and anxiety disorders [14,159,160,161]. There is also some evidence for sex differences in brain activity in ADHD as electroencephalography (EEG) recordings revealed that girls exhibit abnormally elevated coherence in frontal and temporal regions and localised frontal theta enhancement, whereas boys show little evidence of systematic coherence development, and more widespread theta-wave enhancement [162,163,164]. These findings are in line with the notion of a more extensive and severe neurodevelopmental phenotype in males with ADHD. A recent genome-wide association study of ADHD patient samples from Psychiatric Genomics Consortium (PGC) and the Lundbeck Foundation Initiative for Integrative Psychiatric Research (iPSYCH) revealed a greater familial burden of risk in female individuals with ADHD, although autosomal common variants did not explain the sex bias in ADHD prevalence [165]. Overall, sex differences observed in ADHD and autism may be partially accounted for by diagnostic and ascertainment biases, but are likely, in large part, to be due to biological differences between males and females. 

#### 3.1.3. Tourette’s Syndrome

Tourette’s syndrome is a neuropsychiatric disorder characterised by recurring motor and phonic tics during childhood and adolescence [166]. Typical onset of Tourette’s syndrome occurs around 6 to 7 years old and one-third of Tourette’s affected children retain their symptoms into adulthood [167,168]. 90% of Tourette’s patients suffer from comorbid psychiatric conditions including ADHD, obsessive compulsive disorder (OCD), aggression and other impulse control disorders [169,170]. Neuroimaging and post-mortem studies have shown excessive activity and/or innervation of the cerebral cortex and basal ganglia of Tourette’s patients [171], which may reflect dysregulations in the dopaminergic system [172,173,174,175]. Treatment options for Tourette’s syndrome include drug treatments such as dopamine antagonists [176,177] or behavioural approaches such as habit reversal therapy [178,179].

Tourette’s syndrome is more common in boys with sex ratio of four boys to one girl [180], and is diagnosed earlier in males than females [181]. In males, onset of Tourette’s syndrome is characterised by anger-related manifestations and simple tics; conversely, females exhibit complex tics more often than males [181]. Furthermore, male TS patients exhibit significant deficits in cortical and callosal thickness, which are not observed in females [182,183,184]. Whilst males have increased vulnerability for tics in childhood, females have greater tic severity during adulthood [185].

#### 3.1.4. Early-Onset Schizophrenia

Although schizophrenia has a weak male bias on average (7 males: 5 females), this ratio is increased in younger males (<20 years old) where two males to every female are affected [186,187,188,189]. Males also have an earlier age of onset of schizophrenia, between 18–25 years of age, compared with the female age of onset which is 25–35 years [73]. Males tend to have a greater vulnerability to negative symptoms and traits of disorganization, while females more frequently exhibit depressive symptoms [73].

### 3.2. Male-Biased Neurodegenerative Disorders

#### 3.2.1. Parkinson’s Disease

Parkinson’s disease (PD) is the second most common neurodegenerative disorder after AD, affecting nearly 10 million people worldwide. PD affects 2% of population over the age of 65, increasing to 5% over the age of 85 [190]. PD is characterized by the inability to initiate and maintain voluntary movement [190]. Motor symptoms of PD are associated with the loss of midbrain dopamine neurons [191]. Whilst current therapies based on DA replacement strategies effectively treat motor symptoms, they do not slow down or halt the progression of PD. Furthermore, their therapeutic benefit is eventually marred by the development of debilitating side effects known as dyskinesias [192]. Whilst the cause(s) of PD is unknown, PD is likely to arise from a complex interplay between genetic and environmental factors [193,194,195,196].

Aside from aging, the male-sex is the most significant risk factor for PD. Men are twice more likely to develop PD and also have an earlier age of onset and a faster rate of disease progression in PD than women [16,17,197,198]. Evidence from animal models of PD reproduce the sex differences observed in humans, as administration of equal doses of dopamine toxins produce greater motor deficits and nigrostriatal dopamine loss in male rodents and primates than their female counterparts [199,200]. Microarray analysis of single SNc dopamine neurons from healthy post-mortem SNc revealed that expression of genes implicated in PD pathogenesis (e.g., *α-synuclein*, *PINK-1*) was higher in men than in women [201], suggesting that nigral dopamine cells have intrinsic sex differences that may influence the pattern of gene expression, predisposing the male-sex to developing PD. There are also clear differences in expression and function of dopamine machinery genes between males and females. For instance, striatal D2R density and binding potential decline twice as fast with age in males compared to females [202], which is likely to reflect sex differences in symptom severity and response to medication in PD. Together, evidence from animal models and clinical studies suggests that sex differences in PD pathogenesis mechanisms and dopamine machinery genes are likely to contribute to sex differences in prevalence, symptoms severity and medication response in PD. 

#### 3.2.2. Motor Neuron Disease

Motor neuron disease (MND), also known as amyotrophic lateral sclerosis (ALS) or Lou Gehrig’s disease, is a rapidly degenerative muscular disorder affecting over 2.7 per 100,000 people [203]. Although MND has >99% fatality within 2–5 years [204], one of the most famous sufferer Stephen Hawking survived well over 50 years from his initial diagnosis. The pathological hallmark of MND is the presence of hyper phosphorylated and ubiquitinated aggregates of TAR DNA-binding protein 43 (TDP-43), a ubiquitously expressed nuclear protein involved in transcriptional repression and RNA splicing [205,206,207,208]. Whilst the pathogenic role of TDP-43 aggregates remains unclear, it is likely due to either a loss of function, toxic gain of function, or a combination of both [209]. Clinically there is a fast-focal onset but movement impairment eventually spreads all over the body [210], with phenotype heterogeneity attributed to variation in disease progression anatomically. 

Around 95% of MND cases are idiopathic in nature, with the remaining 5–10% of cases attributed to familial genetic origins [211]. Males are at greater risk for MND with a sex ratio of 1.6 male: 1 female for prevalence and 1.4 male: 1 female for incidence [18,19,212,213]. The incidence ratio has been reported to be as high as four males: one female in the 20–29-year age of onset group [214]. Men also have an earlier age of onset for MND than women [215,216,217]. This male-bias is reflected in the superoxide dismutase 1 (SOD1) mouse model of MND as male SOD1 mice had an earlier disease onset than female counterparts [218]. There are sex differences in the clinical features of MND, with men more likely to have spinal onset MND (associated with limb muscle wasting), whereas bulbar onset MND (associated with dysarthria and dysphagia for solids or liquids) are more common in women [18].

## 4. Role of Sex Hormones and Sex Chromosome Genes in Susceptibility to Neurological Disorders

Sex differences in brain and behaviour have been largely attributed to the effects of sex hormones, including permanent or “organizational” effects during development and reversible or “activational” effects during adulthood [219]. However, it is becoming increasingly clear that brain sex differences are also mediated by the complement of genes encoded on the sex chromosomes, which are expressed in a sex-specific manner that is independent of the effects of sex hormones [3,26,37,38]. Indeed, sex chromosome gene expression is sexually dimorphic in the brain in a region-specific, cell type-specific, and time-specific manner [220,221,222,223,224,225,226,227,228]. Here, we will discuss the relative contribution of sex-hormones and sex-chromosome complement genes in the susceptibility to neurodegenerative and neuropsychiatric disorders.

### 4.1. Influence of Sex Hormones 

#### 4.1.1. Estrogen and Estrogen Signalling

Compelling evidence demonstrate that estrogen exerts neuroprotective actions in females [7]. For instance, post-menopausal women also have an increased disease risk of PD and AD as they have the lowest concentrations of circulating estrogen, even lower than males [92]. Similarly, symptoms worsen in female PD or MS patients with the onset of menopause or before the onset of menses during the menstrual cycle [229,230]. Furthermore, early initiation of estrogen replacement therapy at menopause appears to lower the risk of developing AD [79,231], whilst post-menopausal estrogen treatment has been associated with reduced risk of developing PD [232] and reduced symptom severity in female PD patients [233,234]. Likewise, increased levels of estrogen produced during pregnancy are associated with reduced severity of MS [230,235]. Studies in animal models recapitulate these clinical findings as treatment with estrogen or estradiol attenuates neuropathology and symptoms in animal models of PD [199,200,236], MS [237], and AD [238,239]. Estrogen exerts its protective effects by mediating mitochondrial function [240], anti-apoptotic mechanisms [240] and immune responses [230], as well as reducing beta amyloid neurotoxicity [241].

Estrogen also plays a significant role in sex differences in susceptibility, symptom severity and treatment response to affective disorders and neuropsychiatric disorders. Women may have a higher risk for developing anxiety disorders, or exacerbation of their present symptoms, during different phases of their reproductive lives, such as puberty, menses, pregnancy, postpartum, and menopause [242,243,244,245,246], whilst depressed women have significantly lower circulating levels of estrogen [247]. Additionally, depressed women had more favorable response to sertraline than to imipramine and the reverse was found with men [52]. Given that the female response was primarily in premenopausal women, this suggested that female sex hormones may enhance response to SSRIs or inhibit response to tricyclics [248]. Similar to PD, the incidence of schizophrenia in post-menopausal females is higher compared to pre-menopausal females [249]. Women have been shown to be more vulnerable to psychotic breakdown at times of estrogen withdrawal, for example just after giving birth and at menopause [74]. Conversely, increased levels of estrogen during the menstrual cycle is associated with an improvement of schizophrenic symptoms and therapeutic response to treatments [250,251]. In support, a clinical study of female schizophrenic patients demonstrated that estrogen or estradiol treatment with antipsychotic drugs led to significant improvements in acute and severe psychotic symptoms when compared to antipsychotic drugs given alone [252,253]. 

Whilst the effects of estrogen on neurodegeneration and cognition are well established, the roles of the estrogen receptors alpha (ERα) and beta (ERβ) are less clear. In the brain, ERα mRNA is abundantly expressed in the hypothalamus and amygdala [254,255,256], whilst ERβ mRNA is highly expressed in the hippocampus and entorhinal cortex [254,255,256]. In line with the brain distribution pattern, female ERβ knockout (KO) mice exhibited spatial learning and memory deficits compared to wild-type controls [257], which was not observed in the ERα KO mice [258]. Moreover, treatment with the ERβ (but not ERα) agonist in ovariectomized mice improved performance in spatial memory tasks [259], suggesting a role ERβ in memory and cognitive processing. In contrast, ERα, but not ERβ, agonist exerted neuroprotective effects in the 1-methyl-4-phenyl-1,2,3,6-tetrahydropyridine (MPTP)-induced mice model of PD [260]. Furthermore, ERα KO mice exhibited greater vulnerability to MPTP-induced DA depletion compared to WT mice, which was not observed in the ERβ KO mice [261]—suggesting a role for ERα in estrogen-mediated neuroprotection. Taken together, greater understanding of the nature of ER selective ligands and the function of ERα and ERβ subtypes in different brain regions may lead to optimal therapies for neurodegenerative and neuropsychiatric diseases.

#### 4.1.2. Testosterone

Prenatal testosterone plays a crucial role in masculinising the developing male brain [219]. Testosterone activates the androgen receptor to mediate the masculinization of the male brain during the perinatal period to induce male-typical behaviours such as aggression and sexual behavior in male adult rodents [262,263]. Thus, exposure to aberrant levels of prenatal testosterone may influence susceptibility to neurodevelopmental disorders, [264,265]. 

In 2002, Baron-Cohen proposed the so-called “extreme male brain theory” [144] which hypothesizes that abnormally high levels of foetal testosterone may underlie the cognitive and emotional profile of people with autism, as well as the higher prevalence of autism in males compared with females. In one study, researchers found that girls who had been exposed to high levels of foetal testosterone in the womb had a more male-typical play style [266], whilst another study found a positive relationship between fetal testosterone levels and number of autistic traits in children [267]. Along these lines, males diagnosed with ADHD or autism have lower finger-length ratios [35,36], a proxy measure of high levels of prenatal testosterone exposure [268].

On the other hand, testosterone and its metabolites have been shown to possess anxiolytic properties, reducing anxiety behaviors and enhancing cognition in male rodents [269,270,271]. Testosterone has shown to influence the severity of tics, as anabolic androgens worsen tic severity in males with Tourette’s syndrome [272], whilst treatment with the 5α-reductase inhibitor, finasteride, decreased tic severity and compulsive symptoms in adult men with Tourette’s syndrome [273]. Depletion of endogenous testosterone levels in male rodents, via castration, induced neuronal cell loss in animal models of MS [274] and PD [275], suggesting that testosterone may be neuroprotective in males. 

In summary, oestrogen exerts neuroprotective effects in females, which may underlie the reduced (or increased) incidence of neurological disorders in females throughout development and adulthood. Exposure to prenatal testosterone appears to be critical for the masculinization of the male brain and hence exposure to abnormal levels of testosterone may contribute to susceptibility of males to neurodevelopmental disorders such as autism and Tourette’s syndrome.

### 4.2. Influence of Sex Chromosome Genes

In addition to the well-established actions of sex hormones on brain sex differences, accumulating evidence demonstrate that sex chromosomes genes (X-linked or Y-linked) can directly influence brain function [3,26,37,38]. Whilst small in number, X- and Y-linked genes are proportionally abundant in the brain and have been shown to influence neural development and function [276,277]. For instance, genes on the sex chromosome may influence neurological diseases by altering the basic differentiation process of the neurons [278], encoding proteins [279], neurotransmitter biosynthesis [26,280] and synaptic transmission [281]. Moreover, sex chromosome abnormalities can influence neurodevelopment and often result in impairments in attention, working memory, verbal skills and executive function [282]. Therefore, the study of sex chromosomes in brain disorders may provide a new angle to understand the sex differences in the pathogeneses of neurodevelopmental and degenerative diseases. To better understand the relative contribution of sex chromosome complement on sex-bias in preponderance to neurological disorders, we will highlight three distinct genetic mechanisms: (i) X-linked dosage effects, (ii) X-linked imprinting effects, and (iii) Y-chromosome effects.

#### 4.2.1. X-Linked Dosage Effects

As females have two X chromosomes, one of the copies of the X chromosome present in females is inactivated in process known as X-inactivation [283], equalizing the gene products of sex chromosomes between the sexes. However, approximately 15–20% of X-linked genes consistently escape X-inactivation [284], and therefore may be expressed higher in females than males. For instance, X-inactivation gene escapees, *Utx* and *Usp9x*, have higher expression in XX mice brains compared to XY, regardless of their gonadal phenotype [285,286]. Whilst the significance of X-inactivation gene escapee expression in brain function remains to be fully elucidated, it may potentially mask any gain or loss of function in females [9]. For instance, females with Turner syndrome (also known as 45, XO), a condition in which a female is partly or completely missing an X chromosome, have increased vulnerability to neurodevelopmental disorders such as ADHD [287], autism [288] and potentially schizophrenia [289]. Similarly, 39, XO mice (female mice with only one X-chromosome) exhibit attention deficits compared to 40, XX mice [290]. These attention deficits were rescued in 40, XY*^X^ mice (39, XO mice with a small number of pseudoautosomal and X-linked genes on the Y*^X^ chromosome), suggesting a protective role for X-chromosome genes in attentional and cognitive processes [290]. Deletions and frameshift mutations of *NLGN4X*, an X-inactivation gene escapee involved in formation and remodeling of synapses, were identified in boys with autism [291,292,293]. Polymorphism in the promoter region of *MAOA*, a X-linked gene involved in catecholamine metabolism, have been associated with increased risk of males to autism [294,295,296] and ADHD [297]. Thus, the extra dose of X-chromosome may have a protective effect in females, reducing the vulnerability to neurodevelopmental disorders such as autism and ADHD. On the other hand, individuals with an additional X-chromosome (e.g., 47, XXX and 47, XXY) exhibit global intellectual impairment [298] and show increased risk of ADHD and autism [299,300], suggesting that over dosage of X chromosomes can also be detrimental to brain development.

#### 4.2.2. X-Linked Imprinting Effects

Imprinted genes are solely expressed by one allele in a parent-of-origin dependent manner [9,301]. Although small in number, high proportion of imprinted genes are expressed in the brain and postulated to have role in neurodevelopment, brain function and behaviour [276]. Given the unique inheritance pattern of the X chromosome—maternal X can be inherited to male or female offspring, whilst paternal X only goes to female offspring—imprinted genes on the X chromosome could potentially influence sex-bias in vulnerability to neurodevelopmental disorders. For instance, any protective function of paternally expressed X-linked imprinted genes will be passed only to the female offspring. Indeed, females with Turner’s syndrome (45, XO) who inherited the paternal X chromosome (45, X^p^O) had superior verbal and higher-order executive function skills compared to females that inherited the maternal X chromosome (45, X^m^O). In view of these findings, they suggested that a genetic locus for social cognition is imprinted on the paternal X-chromosome, which could contribute to the higher incidence of autism seen in 45, X^m^O females, and males [302]. Bishop, et al. [303] assessed verbal and visuospatial memory in females with a single paternal X chromosome (45, X^p^O) and those with a single maternal X (45, X^m^O). Their findings revealed that 45, X^m^O females showed enhanced verbal forgetting relative to controls, whilst 45, X^p^O females showed disproportionate visuospatial memory loss relative to controls. In view of their results, Bishop and colleagues postulated the existence of one or more imprinted genes involved in memory function on both the maternal and paternal X chromosome Bishop, Canning, Elgar, Morris, Jacobs and Skuse [303], which could contribute to sex differences seen in memory-associated neurological disorders. To better understand the role of X-linked genes in cognitive function, Davies, et al. [304] assessed the performance of 39, XO mice, where the X chromosome was either paternally (39, X^p^O) or maternally (39, X^m^O) inherited, in various cognitive tasks. 39, X^m^O mice exhibited deficits in reversal learning, a measure of impulsive and compulsive behaviour [305]. Furthermore, a novel imprinted gene candidate, *Xlr3b*, which is maternally expressed in the 39, X^m^O mouse prefrontal and orbitofrontal cortex and hippocampus, was identified as mediator of the inflexible reversal learning. Together, these findings indicate that X-linked imprinted genes expressed in a parent of origin-dependent manner could influence sexually dimorphic phenotypes in brain function and may confer vulnerability to neurodevelopmental disorders such as autism and ADHD.

#### 4.2.3. Y-Chromosome Effects 

The Y chromosome is passed only from father to son, indicating that any Y-linked traits are only present in males. Males possess genes on the Y chromosome that have no homologous sequences on the X chromosome, suggesting that genes encoded on the Y chromosome may contribute to biological sex differences [306,307]. Whilst the majority of Y chromosome genes are involved in testis development and spermatogenesis [308,309,310], a significant proportion are expressed in the brain [225,226]. Thus, Y chromosome-specific genes could contribute to sexual differentiation of the brain indirectly, through influencing gonadal hormone production or directly via cell autonomous actions in the brain [277]. Xu and colleagues showed that six Y-chromosome genes, *Ddx3y*, *Ube1y*, *Kdm5d*, *Eif2s3y*, *Uty* and *Usp9y* were expressed in both the developing and adult XY male mouse brain [226]. All six genes were also expressed in the brain of XY female mice that lacked the male-sex determining gene *SRY* and testes [226]. Another study showed that *Dby* and *Eif2s3y* were expressed in the developing male mouse brain at 10.5 days post coitum, prior to the influence of gonadal hormones [38]. Together, these findings indicate that Y-chromosome gene expression in the brain is independent of hormonal influence, suggesting that Y chromosome genes could contribute to sexually dimorphic brain development and function. In support, several clinical studies have reported ADHD diagnosis in 47, XYY and 48, XXYY boys [299,311,312] which suggests that a dosage-effect of Y-chromosome genes may increase the risk of males to neurological disorders. Whilst it is unclear which Y-chromosome gene(s) have a physiological role in the male brain, emerging studies indicate that the male-sex determining gene, *SRY,* is an ideal candidate to investigate [26,280].

#### 4.2.4. Y-Chromosome Gene *SRY*


*SRY* (Sex-determining Region on the Y chromosome) encodes is a transcription factor that initiates male-sex determination by directing embryonic bipotential gonads to develop into testes rather than ovaries [313,314]. Subsequently the testes secrete testosterone, which can act to masculinise the brain during development (‘organisational effects’) or at particular time points leading to reversible neural changes (‘activational effects’). In addition to its ‘indirect’ hormonal effects on the brain, emerging evidence demonstrate that *SRY* can directly exert actions in the adult male brain [26]. In the mouse brain, *SRY* expression is developmentally regulated, with the circular (untranslatable) form of *SRY* transcripts expressed from embryonic day 11 (E11) through E19 whilst postnatal brain *SRY* transcripts are of the linear (translatable) form [315]. In the adult brain, *SRY* mRNA is expressed in regions abundant in catecholamine cell bodies or nerve terminals such as the SNc, ventral tegmental area (VTA), locus coeruleus and hypothalamus [26,316]. Immunohistochemical studies in human and rodent midbrain sections reveal that *SRY* protein co-localizes with dopaminergic neurons in the SNc and VTA, and with GABAergic neurons in the substantia nigra pars reticulata [26,280]. In line with the presence of *SRY* in dopamine-neurons, *SRY* regulates dopamine biosynthesis genes in vitro [280,316,317] and dopamine-dependent functions such as voluntary movement [26] and blood pressure [318,319] in male rats. Combined, these results suggest that *SRY* exerts a direct male-specific action on adult DA neurons, independent of circulating gonadal hormones.

Given that *SRY* is expressed in brain regions closely associated with pathophysiology of ADHD and autism, abnormal regulation of *SRY* expression during development, and consequently dopamine machinery genes, may contribute to the hyper- or hypo-function of dopamine levels in these disorders. Considering the expression of *SRY* in the human male SNc [280], a brain region that degenerates in PD, dysregulation of *SRY* in male dopamine neurons may underlie the male preponderance to PD. Indeed, *SRY* expression is aberrantly elevated in a human cell culture model of PD [320]. Whilst other Y-linked genes, such as neuroligin 4, will also need to be examined for their role in male-biased neurodevelopmental disorders [321], these results highlight the need to better understand the molecular regulation, function, and targets of *SRY* in the healthy and diseased male brain. This information, alongside identifying novel *SRY* polymorphisms, will be essential for the development of novel therapeutic strategies (e.g., male-specific therapies) for sex-biased neurological disorders. 

## 5. Future Directions and Conclusions

Given that sex differences in neurological disorders arises from a combination of hormonal and genetic factors, integrated pre-clinical and clinical research efforts will be needed to distinguish the source of these differences. Animal models in which chromosomal and hormonal factors can be systematically varied, such as the four-core genotype model [37] and sex chromosome trisomy model [322], have proven to be useful tools in partitioning the effect of sex chromosome complement from the actions of gonadal sex hormones. The use of transgenic or knockout rat technologies and spatiotemporal-restricted gene expression strategies (e.g., brain *SRY* overexpressing transgenic rat) will be vital in identifying the role of sex-linked genes in the healthy and diseased brain. In parallel, genome-wide association and copy-number variation studies in human populations need to stratify disease-associated genetic variants by sex to identify potential interactions between sex and genetic vulnerability. Together, these studies may highlight novel sex-specific protective and risk factors, which should ultimately facilitate improved diagnosis, prognosis and treatment for males and females.

Evidence from numerous clinical studies indicate a promising role for sex-specific hormonal therapies in neurodegenerative and neuropsychiatric disorders. For instance, female-specific therapeutic benefit of estrogen treatment has been demonstrated for cognitive decline in AD [323,324,325,326] and MS [327], as well as improved efficacy of dopamine agonist medications for female PD patients [233]. Similarly, estrogen has shown to be beneficial in women with post-natal depression [328] and schizophrenia [34]. Conversely, testosterone therapy has shown promising effects on cognitive decline in male patients with AD [329] or MS [330], and non-motor symptoms of PD [331]. Moreover, emerging evidence from animal models and clinical observations also suggest a role for targeting X-linked (e.g., steroid sulfatase) or-Y-linked (*SRY*, neuroligin-4) genes [26,38,321] for male-biased neurodevelopmental disorders such as autism or ADHD. Given the potential therapeutic benefit of gonadal hormones in neuropsychiatric and neurodegenerative disorders, better understanding the sex hormone status of patients have the potential to improve diagnosis and inform treatment selection.

Whilst gender specific medicine still faces many challenges in current healthcare, the recognition that sex affects the pathophysiology and expression of human disease is starting to influence governmental and regulatory bodies. Indeed, National Institutes of Health and Food and Drug Administration are acting to ensure that both sexes are represented in pre-clinical and clinical studies, as well as all the drug development phases. In conclusion, better understanding the biological differences between the male and female brain will not only allow better, and even optimal, treatment of neurological disorders and a reduction in health care expenditure due to the elimination of inadequate treatment or adverse events. 

## Figures and Tables

**Figure 1 brainsci-08-00154-f001:**
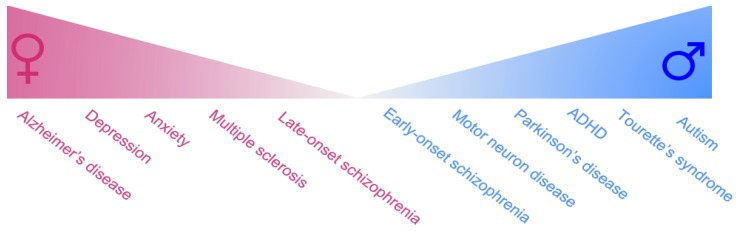
Sex differences in the prevalence of neurodegenerative and neuropsychiatric disorders. Abbreviations: ADHD, Attention-deficit hyperactivity disorder.

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
