# Peer review of "Sex: A Significant Risk Factor for Neurodevelopmental and Neurodegenerative Disorders"

_brainsci, 2018, doi:10.3390/brainsci8080154_

Round 1

Reviewer 1 Report

This review addresses current knowledge and understanding behind sex differences in neurodevelopmental and neurodegenerative diseases. Such sex difference have been known for long, but the underlying molecular mechanisms are yet largely obscure. It is clear that understanding these sex-differences on a cellular and molecular level may bring new knowledge behind neurological diseases where sex is a risk factor, and promote the development of new treatment strategies adapted to men or women. Therefore, the scientific and regulatory communities are rightfully increasing their attention into such sex differences, making this review timely.

The review is well-written, and I only have some minor concerns that I think the authors should address:

Lines 132 – 140: Females have higher prevalence to anxiety and depression. But is there any evidence for a neurological difference underlying the increased prevalence or is it simply due to socilogical factors such as male unwillingness to report their anxiety symptoms? The authors should explain this more clearly.

Sentence, line 158 – 160: The authors should add increased incidence of depression and anxiety also after child birth (post partum depression), which is known to be linked to rapid drop in estrogen levels.

One important piece of information that the authors must include in the section about AD (from line 186) or in the section about estrogen (from line 390) is that early initiation of hormone replacement therapy (e.g. with estrogen) at menopause appears to lower the risk of developing AD (which in turn may imply that menopause in itself is a risk factor for AD). These exciting results come from observational studies, but more epidemiology and experimental research is necessary to establish exactly how sex hormones may be neuroprotective in AD.

Two references on this topic (reviews):

Mielke, M. M., Vemuri, P. & Rocca, W. A. Clinical epidemiology of Alzheimer's disease: assessing sex and gender differences. Clinical epidemiology 6, 37-48, doi:10.2147/CLEP.S37929 (2014).

And:

Merlo S, Spampinato SF, & Sortino MA (2017) Estrogen and Alzheimer's disease: Still an attractive topic despite disappointment from early clinical results. Eur J Pharmacol 817:51-58.

In the section about influence of sex hormones / Estrogen (starting line 390): The authors should briefly introduce  the estrogen receptors alpha and beta, the main mediators of estrogen signalling.  Are there any differences between the isoforms on  neuroprotective and neurocognitive implications of estrogen (e.g. during development)?

Some excellent reviews on the topic: 

Varshney, M. & Nalvarte, I. Genes, Gender, Environment, and Novel Functions of Estrogen Receptor Beta in the Susceptibility to Neurodevelopmental Disorders. Brain sciences 7, doi:10.3390/brainsci7030024 (2017)

And:

Crider, A. & Pillai, A. Estrogen Signaling as a Therapeutic Target in Neurodevelopmental Disorders. The Journal of pharmacology and experimental therapeutics 360, 48-58, doi:10.1124/jpet.116.237412 (2017).

Minor issues

In Figure 1: “Alzheimer’s” should be “Alzheimer’s disease”

Line 131: SSRI and SNRI should be spelled out here and not in lines 150 & 151

Line 233: Typo:” Unaffected female carriers of”... or ... “that carry”...

Line 274: Typo: …“the “extreme male brain” theory testosterone”…

Line 436: Typo: …”influence severity”

Line 439-440: There is a Typo or something is missing in this sentence: “Testosterone may also have neuroprotective effects in males, as castration induced neuronal cell loss and male animal models of MS [264] and PD [265].”

Line 481: There is a Typo or something is missing in this sentence: “…have been associated with increased risk of males to [284-286] and ADHD [287].”

Line 579: There is a typo in this sentence: “genetic variants by sex for to identify potential interactions”

Author Response

Reviewer 1

This review addresses current knowledge and understanding behind sex differences in neurodevelopmental and neurodegenerative diseases. Such sex difference have been known for long, but the underlying molecular mechanisms are yet largely obscure. It is clear that understanding these sex-differences on a cellular and molecular level may bring new knowledge behind neurological diseases where sex is a risk factor and promote the development of new treatment strategies adapted to men or women. Therefore, the scientific and regulatory communities are rightfully increasing their attention into such sex differences, making this review timely.

The review is well-written, and I only have some minor concerns that I think the authors should address:

Lines 132 – 140: Females have higher prevalence to anxiety and depression. But is there any evidence for a neurological difference underlying the increased prevalence or is it simply due to sociological factors such as male unwillingness to report their anxiety symptoms? The authors should explain this more clearly.

Response: We have now explained this more clearly, providing evidence for neurobiological sex differences that underlie the increased prevalence of anxiety in females (line 141 to 144).

Sentence, line 158 – 160: The authors should add increased incidence of depression and anxiety also after child birth (postpartum depression), which is known to be linked to rapid drop in estrogen levels.

Response: We have now added information regarding increased incidence of depression and anxiety following child birth (line 158 to 161).

One important piece of information that the authors must include in the section about AD (from line 186) or in the section about estrogen (from line 390) is that early initiation of hormone replacement therapy (e.g. with estrogen) at menopause appears to lower the risk of developing AD (which in turn may imply that menopause in itself is a risk factor for AD). These exciting results come from observational studies, but more epidemiology and experimental research is necessary to establish exactly how sex hormones may be neuroprotective in AD. Two references on this topic (reviews):

Mielke, M. M., Vemuri, P. & Rocca, W. A. Clinical epidemiology of Alzheimer's disease: assessing sex and gender differences. Clinical epidemiology 6, 37-48, doi:10.2147/CLEP.S37929 (2014).

And:

Merlo S, Spampinato SF, & Sortino MA (2017) Estrogen and Alzheimer's disease: Still an attractive topic despite disappointment from early clinical results. Eur J Pharmacol 817:51-58.

Response: We have now included a sentence describing the beneficial effect of early initiation of estrogen therapy in AD, as well as PD and MS (see line 396 to 399).

In the section about influence of sex hormones / Estrogen (starting line 390): The authors should briefly introduce the estrogen receptors alpha and beta, the main mediators of estrogen signalling.  Are there any differences between the isoforms on neuroprotective and neurocognitive implications of estrogen (e.g. during development)? Some excellent reviews on the topic: 

Varshney, M. & Nalvarte, I. Genes, Gender, Environment, and Novel Functions of Estrogen Receptor Beta in the Susceptibility to Neurodevelopmental Disorders. Brain sciences 7, doi:10.3390/brainsci7030024 (2017)

And:

Crider, A. & Pillai, A. Estrogen Signaling as a Therapeutic Target in Neurodevelopmental Disorders. The Journal of pharmacology and experimental therapeutics 360, 48-58, doi:10.1124/jpet.116.237412 (2017).

Response: We have now added a section on the role of the estrogen receptors alpha and beta, discussing the differences between the isoforms on neuroprotective and neurocognitive effects of estrogen (see line 421 to 434).

Minor issues

In Figure 1: “Alzheimer’s” should be “Alzheimer’s disease”

Response: We have now changed “Alzheimer’s” to “Alzheimer’s disease” and “Tourette’s” to Tourette’s syndrome in Figure 1.

Line 131: SSRI and SNRI should be spelled out here and not in lines 150 & 151

Response: We have now spelled out SSRI and SNRI in full in line 130-131.

Line 233: Typo:” Unaffected female carriers of”... or ... “that carry”...

Response: We have corrected this error in line 232

Line 274: Typo: …“the “extreme male brain” theory testosterone”…

Response: We have now corrected this error in line 273.

Line 436: Typo: …”influence severity”

Response: We have now corrected this error in line 451

Line 439-440: There is a Typo or something is missing in this sentence: “Testosterone may also have neuroprotective effects in males, as castration induced neuronal cell loss and male animal models of MS [264] and PD [265].”

Response: We have now corrected this sentence to “Depletion of endogenous testosterone levels in male rodents, via castration, induced neuronal cell loss in animal models of MS [265] and PD [266], suggesting that testosterone may be neuroprotective in males.” (line 454 to 456)

Line 481: There is a Typo or something is missing in this sentence: “…have been associated with increased risk of males to [284-286] and ADHD [287].”

Response: We have now corrected this sentence to “….have been associated with increased risk of males to autism [284-286] and ADHD [287].” (line 498)

Line 579: There is a typo in this sentence: “genetic variants by sex for to identify potential interactions”

Response: We have now corrected this sentence to “genetic variants by sex to identify potential interactions” (line 597)

Reviewer 2 Report

Please mention extent of sex differences whenever sex differences are stated.

Are you saying that early onset and late onset are two different diseases, one developmental and the other degenerative? 

With current knowledge, how can treatment for neuropsychiatric disease be improved?

Please correct spelling, grammar, capitalization errors.

Author Response

Reviewer 2

Comments and Suggestions for Authors

Please mention extent of sex differences whenever sex differences are stated.

Response: We have now mentioned extent of sex differences whenever such information was available.

Are you saying that early onset and late onset are two different diseases, one developmental and the other degenerative? 

Response:  We are not saying that early onset and late onset Schizophrenia are two different diseases (i.e. one developmental and the other degenerative), but rather different in their susceptibility to males and females (i.e. early-onset being male-biased and late-onset being female-biased). Nonetheless, the reviewer raises an interesting idea that early- and late-onset Schizophrenia may have different pathogenic processes, although further work in pre-clinical animal models and human post-mortem brain samples is needed to confirm this.

With current knowledge, how can treatment for neuropsychiatric disease be improved?

Response: “Given the potential therapeutic benefit of gonadal hormones in neuropsychiatric and neurodegenerative disorders, better understanding the sex hormone status of patients have the potential to improve diagnosis and inform treatment selection”.

This sentence has now been added to line 610 -613.

Please correct spelling, grammar, capitalization errors.

Response: We have now proofed and corrected spelling, grammar, capitalization errors throughout the manuscript.